# Integrative Bioinformatic and Epidemiological Analysis of Acetaminophen Use and Risk of Sex Hormone-Related Cancers

**DOI:** 10.3390/ijms27010376

**Published:** 2025-12-29

**Authors:** Filip Górawski, Zofia Wicik, Kamilla Blecharz-Klin

**Affiliations:** 1Department of Experimental and Clinical Pharmacology, Centre for Preclinical Research and Technology CePT, Medical University of Warsaw, Banacha 1B, 02-097 Warsaw, Poland; 2Department of Experimental and Clinical Neuroscience, Institute of Psychiatry and Neurology, Sobieskiego 9, 02-957 Warsaw, Poland

**Keywords:** paracetamol, acetaminophen, cancer risk, sex hormones, in silico analysis, bioinformatic analysis

## Abstract

Acetaminophen (paracetamol) is one of the most widely used analgesic and antipyretic drugs worldwide, yet its potential impact on hormonal balance and the risk of hormone-dependent cancers remains unclear. This study aimed to integrate epidemiological and bioinformatic evidence to assess the association between acetaminophen use and the risk of sex hormone–related cancers. A systematic review of preclinical and human studies was complemented by in silico analyses of acetaminophen’s molecular targets and their involvement in cancer-related pathways. Epidemiological data indicate that, although experimental studies suggest possible hormonal and reproductive effects, most population-based studies do not support an increased cancer risk, and some even suggest a potential protective effect. Bioinformatic analyses identified genes and pathways associated with ovarian and prostate cancers that may be modulated by acetaminophen, as well as possible links with breast cancer through drug metabolism–related genes. These findings reveal a shared molecular network that may underlie the observed epidemiological patterns. This integrative analysis underscores the need for further basic and clinical research to elucidate acetaminophen’s role in hormone-related carcinogenesis and to inform its safe therapeutic use.

## 1. Introduction

Acetaminophen, also known as paracetamol, is one of the most commonly used analgesics and antipyretics globally. Available over the counter, it is frequently chosen for treating mild pain and is also used in combination with more potent opioids in multimodal cancer pain management [1]. This drug may not only be an element of cancer pain therapy but also have potential protective effects and modulate pathways involved in the development of some types of cancer [2]. On the other hand, some researchers present the opposite view, suggesting that acetaminophen is a potential suppressor of antitumor immunity and may increase the risk of liver and renal carcinogenesis as well as some forms of lymphohematopoietic cancer [3,4,5]. A comprehensive study, including data from the US Kidney Cancer Study, the Prostate, Lung, Colorectal, and Ovarian Cancer Screening Trial, and a meta-analysis, found that over-the-counter acetaminophen use is associated with a higher risk of kidney cancer, with an odds ratio (OR) of 1.35 (95% CI = 1.01–1.83). Risk increased with the duration of use, with a twofold higher risk for those using it for 10 years or more (OR = 2.01; 95% CI = 1.30–3.12). The study also noted a link between acetaminophen use and the increased risk of renal cell carcinoma (hazard ratio HR = 1.68; 95% CI = 1.19–2.39), although the risk did not increase among a few long-term users. The meta-analysis also confirmed this association, indicating an increased risk of kidney cancer related to acetaminophen use. In cohort studies, the pooled relative risk (RR) for renal cancer after acetaminophen use was 1.34 (95% CI = 1.13–1.59), and in case–control studies, 1.20 (95% CI = 1.01–1.42) [6].

These scientific findings remain inconclusive and often contradictory. In this review, we attempt to answer whether acetaminophen may exert endocrine-related effects and whether it could influence the risk of sex hormone-dependent cancers. This issue is particularly important because acetaminophen is considered one of the safest medications for various groups of patients, especially those who cannot use nonsteroidal anti-inflammatory drugs (NSAIDs), as well as in sensitive stages of human development such as pregnancy, breastfeeding, or the early postnatal period [7]. Despite this, its safety during pregnancy is currently being re-evaluated by the scientific community, which results, among other things, from the possible hormonal effect on the fetus and the potential long-term consequences for offspring health [8,9,10]. It remains unclear whether such developmental exposure may influence cancer risk later in life.

Preclinical studies indicate that acetaminophen may negatively affect the hypothalamic-pituitary-gonadal (HPG) axis—a signaling pathway that controls the reproductive system and sexual development of both males and females. The drug has been shown to potentially disrupt HPG axis function and sex hormone secretion, possibly through alterations in hypothalamic neurotransmitters, such as dopamine and norepinephrine, and through changes in the concentration of glutamic acid in the hypothalamus [11,12,13].

Lecante and coworkers suggest that acetaminophen disturbs ovarian steroidogenesis and has significant toxic effects on fetal ovaries without affecting prostaglandin or inhibin B production. Exposure of ovarian explants to acetaminophen between 7 and 12 days post-fertilization resulted in a significant reduction in the percentage of proliferating cells. The findings indicated that at a concentration of 10^−3^ M, acetaminophen led to a 67% decrease in the cell count. The authors demonstrated that therapeutic levels of acetaminophen could interfere with human ovarian development by dysregulating sex steroid production. However, it has not yet been established whether such changes influence long-term reproductive health or cancer risk [14].

It is postulated that acetaminophen might cause endocrine disorders due to its structural similarity to endocrine-disrupting compounds [15]. Some data indicate that the drug changes the hormonal balance, which may raise concerns about its impact on reproductive organs and fertility [16]. Nevertheless, such endocrine-disrupting properties have not been sufficiently confirmed in human populations.

Hurwitz et al. conducted an analysis to investigate the relationship between analgesic use and the level of serum sex hormones in 1860 postmenopausal women. Results of this Women’s Health Initiative Observational Study contradict the hypothesis that acetaminophen affects sex hormone levels. Regular acetaminophen use was not associated with concentration changes in estrogens, androgens, and their metabolites in the human serum [17].

Assuming that acetaminophen can indeed affect the level of sex hormones, the next question is whether the use of the drug can affect the risk of sex hormone-dependent cancers.

To explore this issue, we conducted an integrative review of preclinical and clinical data on the endocrine-related effects of acetaminophen and their potential implications for hormone-dependent carcinogenesis. Additionally, we performed an exploratory in silico bioinformatic analysis to identify candidate genes and pathways potentially linking acetaminophen targets with sex hormone-related cancers. Computational analysis is a useful tool for generating hypotheses regarding drug safety and identifying molecular interactions potentially relevant for clinical outcomes. This approach offers a cost-effective method to explore possible links between acetaminophen exposure and endocrine-related cancer risk.

## 2. Results

### 2.1. Acetaminophen and Male Sex Hormones—Summary of Existing Evidence

The impact of acetaminophen on male sex hormones has been the subject of numerous studies, leading to often contradictory conclusions. To date, most of the evidence derives from preclinical models, and a definitive consensus on the effect of acetaminophen on androgen production and related receptors in humans has not been established. Nevertheless, several experimental studies provide insights into its possible endocrine-disrupting properties.

In a metabolomic analysis conducted by Cohen et al., significant effects of acetaminophen on human androgen metabolism were noted. The study reported that acetaminophen affects sulfated metabolites of steroid hormones—such as androstenediol, pregnenolone, and dehydroepiandrosterone (DHEA)—by reducing levels of neurosteroids like pregnenolone sulfate and dehydroepiandrosterone sulfate (DHEAS), which may in turn influence the secretion of sex hormones [18].

Oskarsson and colleagues confirmed that acetaminophen significantly influences levels of androgens such as DHEA and androstenedione in the human adrenocarcinoma cell line, H295R. The study found that the drug, at a concentration of 1 mM, resulted in a 40% and 20% decrease in DHEA and androstenedione secretion, respectively, compared to control levels [19].

However, a more recent study by Johnson-Ferguson et al. found no significant association between acetaminophen use and the level of steroid hormones in the hair of 1002 young adults, although a potential corticosteroid-inducing effect was noted [20]. These discrepancies may result from differences in study design, dose, exposure duration, and biological matrices analyzed.

A study by Albert et al. demonstrated that mild analgesics can cause hormonal disruptions in the testes. The authors reported that acetaminophen disrupts testosterone production by affecting Leydig cells in vitro. In the cited study, acetaminophen had the weakest effect on androgen secretion among the tested molecules, but the authors observed reduced prostaglandin secretion, antiandrogenic effects, and decreased steroidogenesis. Moreover, acetaminophen inhibits the activity of the enzyme CYP17A1, which is crucial for androgen synthesis [21]. In the study by Van Den Driesche et al., involving the transplantation of human testicular fragments into rats, acetaminophen at a dose of 20 mg/kg administered three times a day for 7 days significantly reduced serum testosterone levels (by 45%) and the mass of seminal vesicles (by 18%). The authors found inhibition of fetal testosterone production after weekly exposure to acetaminophen in a xenograft model. At the same time, they observed reduced expression of key steroidogenic enzymes, including CYP11A1 and CYP17A1. These findings suggest a potential suppressive effect of acetaminophen on steroidogenesis in fetal tissues, although the translational relevance of the xenograft model to human development remains uncertain [22].

Prenatal studies in rats by Pereira et al. describe alterations in testosterone levels in male offspring of mothers exposed to acetaminophen (350 mg/kg/day). Maternal treatment during gestation caused significant changes in testis weight and histomorphometry in adult male offspring. The authors suggest that altering testosterone secretion may influence the activity of the hypothalamic-pituitary-gonadal axis through the use of acetaminophen. Additionally, the possibility of impaired hypothalamic sexual differentiation was raised, potentially affecting adult sexual behavior [23].

In our earlier study, a significant reduction in testicular testosterone levels was observed in rats treated with acetaminophen compared with controls. Gene expression analysis revealed upregulation of several key steroidogenic and gonadotropic factors, including LH-R, FSH-R, GnRH-R, SR-B1, 3β-HSD, 17β-HSD, P450scc, StAR, and the androgen receptor. These changes suggest that acetaminophen may interfere with steroid hormone biosynthesis by affecting both cholesterol transport and enzymatic pathways involved in testosterone production.

The increased expression of gonadotropin receptors and LH-β subunits may represent a compensatory mechanism in response to declining androgen levels, potentially leading to dysregulation of the steroidogenic machinery. The observed gene expression alterations appear to be coordinated and affect multiple levels of the reproductive axis. However, these findings are derived from animal models, and their direct relevance to human reproductive physiology remains to be clarified [24].

Some research has linked acetaminophen exposure with decreased semen quality. Studies in animal models demonstrated that both therapeutic and high doses of acetaminophen led to deteriorated semen parameters, including reduced sperm motility and morphology. Sperm chromatin condensation and DNA fragmentation were observed following short- and long-term administration of acetaminophen, suggesting potential genotoxic effects during spermatogenesis [25]. Other in vivo studies showed that high doses of acetaminophen significantly reduce sperm motility and viability while increasing sperm damage [26]. In human studies, men with higher urinary acetaminophen levels were found to have reduced sperm counts and motility compared with those with lower levels [27].

Overall, the majority of preclinical studies indicate that acetaminophen may interfere with androgen secretion, testicular steroidogenesis, and semen quality. However, clinical evidence in humans remains sparse and inconsistent. Further well-designed epidemiological and experimental studies are needed to clarify the relevance of these findings for human male reproductive health.

#### 2.1.1. Acetaminophen and Androgen-Dependent Cancers—Overview of Current Evidence

##### Acetaminophen Exposure and Prostate Cancer Risk

Chronic inflammation, resulting from hormonal changes, infections, trauma, urine reflux, or dietary factors, is considered a key mechanism in the development of prostate cancer. This has led to the hypothesis that anti-inflammatory drugs, by attenuating the inflammatory response and lowering serum prostate-specific antigen (PSA), could play a protective role [28,29]. In contrast to NSAIDs, acetaminophen has a very weak anti-inflammatory effect, which is negligible in clinical practice.

The relationship between the use of acetaminophen and the occurrence of prostate cancer—the most common malignancy among men—is not entirely clear. The results of the studies included in the analysis are presented in Appendix A.

Data from the National Health and Nutrition Examination Survey suggest a potential association between its regular use and lower PSA levels in men over 40 years of age (0.76 times lower than in non-users), though this finding was not statistically significant [29]. This trend may indicate subtle biological effects of acetaminophen on prostate-related biomarkers and warrants further investigation.

Epidemiological studies examining the relationship between acetaminophen and prostate cancer risk, however, present conflicting results. Some analyses show no association between acetaminophen or most NSAIDs and prostate cancer incidence, with aspirin being the only drug consistently linked to a reduced risk, particularly with long-term or low-dose use [30].

On the other hand, some observational studies have suggested a potential protective association between long-term acetaminophen use and prostate cancer risk. In a large prospective cohort study involving 78,485 men aged 50–76 years, Jacobs et al. observed that participants who reported taking 30 or more acetaminophen tablets per month for five or more consecutive years had a 38% lower risk of total prostate cancer (RR = 0.62; 95% CI: 0.44–0.87) and a 51% lower risk of aggressive prostate cancer (RR = 0.49; 95% CI: 0.27–0.88) compared with non-users [31]. Shorter durations of use (<5 years) were not associated with a significant change in risk.

Similarly, in a case–control study using data from the UK General Practice Research Database, García Rodríguez and González-Pérez analyzed 339,462 men and found that regular acetaminophen use for one year or more was associated with a 35% reduction in prostate cancer risk (OR = 0.65; 95% CI: 0.54–0.78), while short-term use (<1 year) showed no protective effect and was even associated with a slight increase in risk, possibly due to protopathic bias [32]. Despite differences in study design and exposure definitions, both studies support a potential inverse association between long-term, high-frequency acetaminophen use and prostate cancer incidence. However, the observational nature of the data limits causal interpretation, and further studies are needed to clarify the biological plausibility of these findings and their implications for chemoprevention.

Contradictory findings were reported in the Finnish Prostate Cancer Screening Trial, which included 78,615 men undergoing prostate cancer screening or in the control arm. The study found that current users of acetaminophen, identified through prescription records and self-reported over-the-counter use, had an increased risk of prostate cancer, particularly metastatic disease. However, no significant association was observed among past users. The hazard ratios for current acetaminophen users were elevated in both the screening arm (HR = 2.41 for metastatic cancer) and the control arm (HR = 3.44 for metastatic cancer) [33]. The study did not specify exact dosages but highlighted that the increased risk was linked to current, rather than previous, use. These findings suggest that the increased prostate cancer risk may be due to symptoms or conditions leading to acetaminophen use, rather than a direct carcinogenic effect of the drug itself.

Furthermore, other prospective studies, such as the VITamins And Lifestyle (VITAL) cohort, did not confirm a relationship between acetaminophen intake and overall cancer risk, including prostate cancer [3,34,35]. Subgroup analyses in this study only suggested a possible, but statistically nonsignificant, reduction in aggressive prostate cancer risk among high acetaminophen users [35].

Likewise, data from a large, population-based case–control study (the ProtecT study) do not support the hypothesis that nonsteroidal anti-inflammatory drugs and acetaminophen play an important role in prostate cancer etiology and can influence the risk of PSA-detected prostate cancer. Evidence from studies shows only a weak, not statistically significant positive association between acetaminophen use and the risk of prostate cancer (OR = 1.20; 95% CI = 0.90–1.60) [36].

Different conclusions come from preclinical studies. An animal investigation study proved that perinatal exposure to mixtures of anti-androgenic chemicals, which also contained acetaminophen, led to lesions in the rat’s prostate. Sex hormones play a crucial role in the development of the prostate gland, and perinatal disturbances in this area can initiate prostate tumorigenesis. It has been demonstrated that both in the normal prostate gland and prostate cancer, activation of the androgen receptor (AR) promotes prostate growth. The prevalence of prostate cancer may be induced in the prenatal period by different disrupting compounds, especially those that cause elevated levels of estrogens. Such chemicals can cause morphological changes and hyperplasia of the prostatic epithelium [37]. In vitro studies showed that, unlike ibuprofen, acetaminophen did not reduce the survival of human androgen-sensitive and androgen-insensitive prostate cancer cells [38].

Epidemiological studies confirm an association between shortening of anogenital distance and increased risk of prostate cancer [39]. Acetaminophen has been shown to shorten anogenital distance, which supports the hypothesis of its estrogenic properties [40]. A study by Isling et al. confirms that exposure to anti-androgens during early prostate development may increase the risk of pre-cancerous prostate lesions [41]. Animals exposed to the chemicals from gestational day 7 to postnatal day 22, in early adulthood, demonstrated slightly reduced weights of ventral prostate and a statistically significant reduction in mRNA levels for androgen-regulatory genes, e.g., prostate-binding protein subunit C3 (Pbpc3). Alterations in the mRNA level of Pbpc3 can be the result of interaction with androgen/estrogen receptors or interference with the synthesis of steroid hormones. Persistent effects were observed on the rat prostate, suggesting that, also in humans, perinatal exposure to acetaminophen should be considered one of the factors that predispose to the occurrence of prostate cancer [38].

In summary, current evidence does not conclusively establish whether acetaminophen affects prostate cancer risk. The role of acetaminophen as a putative endocrine disruptor in prostate cancer pathogenesis remains unclear.

##### Acetaminophen and Testicular Cancer

This part of the review focuses on publications attempting to investigate the impact of acetaminophen on the occurrence of testicular cancer, which predominantly originates from primordial germ cells, which in males differentiate into sperm during spermatogenesis. Among germ cell tumors, seminomas and non-seminomatous germ cell tumors (NSGCTs) are distinguished, including embryonal carcinoma, yolk sac tumor, teratoma, and choriocarcinoma [42]. It primarily affects young males, and current treatment methods are effective due to the use of appropriate chemotherapy regimens. Given that the cancer affects young individuals, it draws the attention of researchers seeking factors that influence its development.

The most established factor associated with testicular cancer is cryptorchidism [43]. Prolonged use of acetaminophen has been linked to an increased risk of undescended testicles [44].

Male reproductive competencies are extremely driven by androgens like testosterone. Previously published research indicates that in rats, acetaminophen exposure during pregnancy and early life stages causes a reduction in the testicular level of testosterone with simultaneous over-expression of genes important for the maturation and correct functioning of the reproductive system in males [45].

Ex vivo studies by Kristensen et al. demonstrated that testosterone production in fetal rats’ testes was inhibited by acetaminophen at doses from 0.1 µM to 100 µM, reducing testosterone levels by 10% to 50% compared with controls. This suggests a significant impact of acetaminophen on Leydig cells responsible for testosterone production [46]. These findings are corroborated by a xenograft model of human fetal testes transplanted into rats. Exposure to acetaminophen for 7 days resulted in a 45% reduction in testosterone and an 18% reduction in seminal vesicle weight (a biomarker of androgen exposure) [22]. Such significant reductions in testosterone levels and associated hormonal disturbances may additionally suggest a potential impact of acetaminophen on testicular cancer development.

Fisher et al. described studies on the impact of fetal exposure to acetaminophen on the occurrence of testicular dysgenesis syndrome. The authors suggest a significant association between maternal acetaminophen use during weeks 8 to 14 of pregnancy and a measurable reduction in anogenital distance in male infants (95% CI = 0.06–0.49). On average, the reduction corresponded to a small but statistically meaningful decrease compared to the typical values observed in the population. A shortened anogenital distance is a marker of reduced androgen exposure during the male programming period, which the authors directly link to impaired differentiation and growth of male genitalia and, consequently, associate with testicular dysgenesis, cryptorchidism, infertility, and low testosterone levels [40]. In 2013, Gumińska et al. conducted studies aimed at correlating testicular dysgenesis with the occurrence of testicular cancer. The results indicate that cancerous changes appeared only in testes with impaired spermatogenesis due to testicular dysgenesis [47].

Although the number of studies directly examining acetaminophen’s impact on testicular cancer is limited, existing evidence suggests possible causal relationships that warrant further large-scale population research.

### 2.2. Acetaminophen and Female Sex Hormones—Critical Appraisal of the Literature

Hormonal regulation is fundamental to the physiology of the female reproductive system and mammary glands. Imbalances characterized by elevated levels of gonadotropins, estrogens, and androgens, concomitant with reduced progesterone levels, markedly increase the risk of malignancies in the uterus, ovaries, and mammary glands, particularly in the presence of mutations in critical oncogenes or tumor suppressor genes. Female sex hormones govern essential processes such as cellular proliferation, differentiation, and apoptosis within tissues, including the uterus, ovaries, and breasts. Experimental data demonstrate that hormones function as modulators in the pathogenesis of endometrial, ovarian, and breast cancers [48]. While hormones do not serve as initiators of carcinogenesis, they may act as promoters (e.g., estrogens, androgens, and pituitary gonadotropins) or inhibitors (e.g., progesterone) of tumor progression. Elevated levels of gonadotropins, estrogens, or androgens are associated with increased oncogenic risk, especially when coupled with mutations in proto-oncogenes or tumor suppressor genes. Progesterone exerts antiproliferative effects and may confer protective properties against neoplastic progression [49]; however, decreased progesterone levels are insufficient to mitigate the tumor-promoting effects of elevated estrogens, androgens, and gonadotropins. Additionally, hypothalamic gonadoliberin has been shown to inhibit cancer cell proliferation and concurrently safeguard against apoptosis. Emerging evidence suggests that numerous reproductive disorders prevalent among women may originate from aberrant programming during fetal development.

Several epidemiological investigations suggest that regular use of analgesics might reduce the incidence of ovarian and breast cancers, although the data remain inconclusive [50,51]. In the Nurses’ Health Study, serum samples from 740 postmenopausal women utilizing analgesics (aspirin, NSAIDs, or acetaminophen) over two years were analyzed. The findings revealed an inverse correlation between analgesic usage frequency and serum concentrations of estradiol, free estradiol, estrone sulfate, and the estradiol/testosterone ratio. Overall, regular analgesic users exhibited lower circulating levels of sex steroid hormones compared with non-users, potentially contributing to a reduced risk of hormone-dependent cancers in this population.

The proposed chemopreventive effects of anti-inflammatory agents may be mediated through reductions in estrogen levels observed in regular analgesic users. Gates et al. delineated a hormone-mediated mechanism whereby most analgesics decrease concentrations of sex steroid hormones—including estradiol, free estradiol, estrone sulfate, and the estradiol/testosterone ratio—in postmenopausal women [52].

Experimental data from human placental JEG-3 cell studies by Addo et al. demonstrate that acetaminophen significantly downregulates the expression of CYP19A1 (aromatase), a key enzyme in estrogen biosynthesis, by 30% at 0.1 mM and by 41% at 1 mM concentrations. Concomitantly, the researchers observed a 25% reduction in estradiol levels at 1 mM acetaminophen concentration [53].

Given its modulatory effects on hormonal homeostasis, combined with moderate anti-inflammatory properties, acetaminophen is under investigation for its potential impact on hormone-dependent malignancies in women.

#### 2.2.1. Acetaminophen and Estrogen-Dependent Cancers

##### Acetaminophen and Ovarian Cancer

Ovarian cancer is regarded as one of the most lethal gynecological malignancies. The primary challenges in effective treatment arise from late-stage diagnosis and the rapid development of chemotherapy resistance [54]. Estrogens have been implicated in the initiation and progression of ovarian cancer. As outlined in the previous section, acetaminophen may influence levels of female sex hormones, and some studies suggest a potential protective effect regarding ovarian cancer incidence. An extended description of the results taken into account in the analysis of the studies can be found in Appendix A.

However, several investigations report no significant impact of acetaminophen on ovarian cancer risk. Prospective analyses involving 197,486 women from the Nurses’ Health Study (NHS) and Nurses’ Health Study-II (NHS-II) revealed no convincing evidence of an association between acetaminophen use and ovarian cancer incidence, with a hazard ratio (HR) for regular versus no use of 1.14 (95% CI = 0.92–1.43) [55]. These findings align with those of Fairfield et al., who analyzed data from 76,821 NHS participants and found no association (the multivariable relative risk (RR) for using paracetamol on ≥5 days per month was 0.81 (95% CI: 0.46–1.43)) [56]. Trabert et al. similarly reported that acetaminophen use was not associated with a reduced ovarian cancer risk (odds ratio [OR] = 0.99; 95% CI = 0.88–1.12), contrasting with the risk reduction observed for aspirin and NSAIDs [57].

Large cohort studies with 121,700 and 116,429 women also found no evidence linking acetaminophen to increased ovarian cancer risk (HR = 1.02; 95% CI = 0.86–1.21) [58]. Similarly, Lacey et al., studying participants of the Breast Cancer Detection Demonstration Project, reported no effect of acetaminophen on ovarian cancer development (RR = 1.0) [59].

In contrast, a meta-analysis by Bonovas et al. suggested a protective effect of acetaminophen, with regular use associated with a 30% reduction in ovarian cancer risk compared to non-use (random effects RR = 0.70; 95% CI = 0.51–0.95), potentially due to the drug’s mild anti-inflammatory properties [60]. Given the well-established role of inflammation in carcinogenesis, the anti-inflammatory action of acetaminophen and NSAIDs may contribute to decreased cancer incidence [61]. Baandrup et al., in a Danish case–control study of 2.7 million women, found that acetaminophen use correlated with reduced odds of ovarian cancer (OR = 0.82; 95% CI = 0.74–0.92) [62]. A subsequent study by the same author confirmed an 18% risk reduction for epithelial ovarian cancer (OR = 0.45; 95% CI: 0.24–0.86) [63]. Moreover, Schildkraut et al., analyzing data from the North Carolina Ovarian Cancer Study, reported a non-significant trend toward decreased risk associated with acetaminophen use (OR = 0.78; 95% CI = 0.56–1.08) [64].

Conversely, a meta-analysis by Trabert et al., encompassing 13 prospective studies with 758,829 women, found no significant association between frequent acetaminophen use and ovarian cancer risk (HR = 1.05; 95% CI = 0.88–1.24) [65]. Ammundsen et al. further contradicted protective claims, showing no reduced risk in acetaminophen users within the Danish MALOVA study cohort [66].

Regarding survival, data are similarly inconclusive. A large pooled analysis of 12 studies by Dixon et al. indicated no association between regular acetaminophen use and overall survival in ovarian cancer patients (pooled HR = 1.01; 95% CI = 0.93–1.10), with no dose–response relationship observed [67]. Similarly, a study of 1305 women with invasive ovarian cancer found no statistically significant survival benefit among acetaminophen users (HR = 0.91; 95% CI = 0.69–1.20) [68].

In summary, current evidence remains inconclusive regarding acetaminophen’s protective role against ovarian cancer, underscoring the need for further large-scale, dedicated investigations.

##### Acetaminophen and Endometrial Cancer

Endometrial growth is regulated by estrogens produced primarily by the ovaries, which exert mitogenic effects on epithelial and stromal cells during the follicular phase. Endometrial cancer ranks among the most common cancers in women and is histologically heterogeneous, with estrogen dependence serving as a key classification factor. The rising global incidence of this disease [69] has spurred investigations into potential etiologic and modifiable risk factors.

Holinka et al. reported that acetaminophen inhibits estrogen-stimulated alkaline phosphatase activity in endometrial adenocarcinoma cells in a concentration-dependent manner, despite lacking direct binding affinity for estrogen receptors ERα or ERβ [70,71]. This suggests weak antiestrogenic properties of acetaminophen, which may confer a protective effect.

Nonetheless, observational studies consistently fail to support any association between acetaminophen use and endometrial cancer risk. Moysich et al. conducted a case–control study involving 427 women with newly diagnosed endometrial cancer and found no chemoprotective effect [72]. Prospective data from 82,971 women, including 747 endometrial cancer cases, corroborated this lack of association [73]. Population-based analyses by Bodelon et al. (410 women) [74] and Neill et al. (1398 cases and 740 controls) similarly excluded a significant relationship between acetaminophen use and endometrial cancer risk [75].

A meta-analysis by Ding et al. (2017), pooling seven observational studies with 3874 endometrial cancer cases, stratified by BMI, age, and hormone therapy, also found no association between acetaminophen use and cancer risk [76]. Recent analysis by Webb et al. reaffirmed these conclusions [77]. A summary of the results from the cited works is provided in Appendix A.

Overall, evidence does not support a chemoprotective or causative role of acetaminophen in endometrial cancer.

##### Acetaminophen and Breast Cancer

Breast cancer is the most common malignant neoplasm in women (about 20% of cancer cases). Although the causes of this disease are unknown, many factors are known to increase the risk of developing it. Risk factors include the long duration of natural hormonal activity, as well as the intake of drugs that contain sex hormones. As confirmed by numerous studies, sex hormones, especially estrogen, may contribute to the development of tumors, e.g., breast cancer.

The physiological action of estrogens on the mammary gland and its estrogen receptors is stimulation of the duct epithelium and lobules to grow. Neoplastic cells, as well as normal cells, express estrogen receptors. In untransformed cells, the density of estrogen receptors is low, while breast cancer cells overexpress them, suggesting that estrogens are essential for rapid tumor growth. As with other hormone-dependent cancers, overexposure to estrogens in breast cancer may also carry a risk of its development.

Epidemiological studies in pre- and postmenopausal women indicate that acetaminophen use does not significantly affect breast cancer incidence or recurrence risk [78,79,80,81,82]. Cohort studies conducted in Denmark, the USA, and other countries confirmed the lack of a significant association between acetaminophen use and breast cancer risk, although some analyses suggest a possible effect on estrogen metabolites in premenopausal women, which may have genotoxic implications [83]. In contrast, regular use of certain NSAIDs, particularly COX-2 inhibitors and aspirin, shows a moderate protective effect, reducing the risk of breast cancer and its invasive forms [84,85,86,87]. This effect is associated with the downregulation of aromatase expression, a key enzyme in estrogen biosynthesis, and the inhibition of proliferation in hormone-dependent cells [84]. However, not all epidemiological studies are consistent; in some analyses, long-term aspirin use was associated with a decreased risk of ER/PR-negative breast cancer subtypes [85], whereas acetaminophen and other NSAIDs showed no significant protective effect [81,86]. A summary of the study’s results was provided in Appendix A.

Experimental and cellular studies provide additional insights into the mechanisms of acetaminophen and NSAID action. In estrogen-dependent breast cancer cell lines, such as T47D and MCF7, acetaminophen can stimulate cell proliferation via estrogen receptors, and this effect depends on the drug isomer [88,89]. Molecular mechanisms include induction of c-myc RNA expression and NF-κB activation, independently of estrogen receptor presence [90]. Conversely, in triple-negative breast cancer cell models, acetaminophen reduces cell migration and increases sensitivity to chemotherapy, which is related to the modulation of microRNA expression involved in epithelial-to-mesenchymal transition [91]. Studies using the MDA-MB-231 cell line also demonstrated that acetaminophen can inhibit tumor growth in animal models and induce differentiation of cancer stem cells, suggesting potential therapeutic applications [92].

However, results from animal studies are inconsistent and cannot be directly extrapolated to human populations. In studies with young female C57B/6 mice, acetaminophen (100, 200, and 250 mg/kg b.w., i.p., for 3 days) did not elicit a typical estrogenic response, although at the highest dose it showed a limited ability to antagonize estradiol-induced regulation of uterine and hepatic progesterone receptors [93]. Other experimental models observed both estrogen-like and anti-estrogenic effects, depending on dose, exposure duration, and tissue type [71,94]. These discrepancies highlight that animal experiments cannot serve as conclusive evidence of acetaminophen’s effects in humans and emphasize the need to interpret results within the context of the biological model used.

In summary, the effects of acetaminophen on breast cancer risk are complex and context-dependent. In most epidemiological studies, acetaminophen does not show a significant protective or risk-enhancing effect [78,79,80,81,82], although it may modulate estrogen metabolites and influence tumor cell proliferation in experimental models [83,88,89,90]. NSAIDs, especially COX-2 inhibitors, exhibit moderate protective effects through the reduction in estrogen biosynthesis [84,85,86,87]. It should be emphasized that the lack of consistent population-level results prevents the definitive rejection of the hypothesis that acetaminophen affects breast cancer development, highlighting the need for further large-scale epidemiological studies considering population diversity, duration and dosage of therapy, and interactions with hormonal factors to determine potential benefits or risks associated with acetaminophen use in breast cancer prevention or treatment [78,79,80,81,82,83,84,85,86,87,88,89,90,91,92,93,94,95,96].

Therefore, also in the case of breast cancer, no final conclusions can be drawn because neither clinical nor preclinical studies are consistent.

### 2.3. Bioinformatic Results

The possibilities offered by bioinformatics analysis were used to identify the top cancer-related targets interacting with acetaminophen targets. To investigate the genes that are closest neighbors to acetaminophen-targeted genes, we utilized gene interaction analysis. From the human interactome, we extracted 24 direct targets of acetaminophen based on the Binding database and extended the network by its neighbors. Additionally, we selected genes associated with at least five selected cancer-related phenotypes and interacting with acetaminophen targets. Additional visual mapping helped identify specific types of cancers (Figure 1).

As top acetaminophen targets interacting with cancer-related genes, we identified the *PTGS2* gene that encodes cyclooxygenase 2 (COX-2), an inducible isoform that catalyzes the committed step in prostaglandin synthesis; genes *CYP3A4*, *CYP1A1*, and *CYP1A2* encode the cytochrome P450 superfamily of enzymes; and gene *CA9* encodes carbonic anhydrase IX. We identified *ESR1*, *IL6*, *IL1B*, *TP53*, and *KDR* as top cancer-related targets interacting with acetaminophen targets.

In our study, we applied Gene Set Enrichment Analysis (GSEA) to predict cancer phenotypes, rare phenotypes, tissues, pathways, and metabolites associated with acetaminophen activity. This kind of analysis identifies whether predefined gene sets, such as pathways or biological processes, are significantly enriched in a given gene list. Tools like EnrichR compare input genes to curated databases (e.g., KEGG, GO) to reveal biological relevance. In our study, the input lists consisted of predicted target genes and interaction networks for acetaminophen. Enrichment was calculated using the hypergeometric test (equivalent to Fisher’s exact test), which assesses whether the overlap between the input gene set and predefined gene sets from each database is greater than expected by random chance, using the complete human genome as background. The resulting *p*-values were corrected for multiple testing using the Benjamini–Hochberg false discovery rate (FDR) procedure, and results with an adjusted *p* ≤ 0.05 were considered statistically significant. Advantages include improved biological interpretation, reduced noise by analyzing gene sets instead of individual genes, efficient handling of large datasets, and facilitating hypothesis generation for further research. In our study, we analyzed four sets of genes: acetaminophen targets obtained from the binding database, first-level interactors of these targets, genes associated with drug metabolism, and first-level interactors of these genes.

Analysis of sex cancer-related phenotypes pointed out the strongest effect of acetaminophen targets on ovarian cancer and their interaction with prostate cancer. Metabolism-related genes were associated with breast cancer, and their network was associated with prostate cancer. Cell-type analysis highlighted the extended target network on breast cancer cell lines and the immune system, while extended metabolism-associated genes were enriched for prostate gland cancer cells (Figure 2).

The analysis of disease phenotypes revealed an enrichment of acetaminophen-related genes in inflammatory breast cancer, as well as an association between acetaminophen metabolites and testotoxicosis. Analysis of MAGMA drugs and diseases pointed out gestational choriocarcinoma for acetaminophen first-level interactors and ovarian and testicular cancer for an extended network of acetaminophen metabolites (Figure 3).

Analysis of pathways pointed to the strongest effect of acetaminophen targets on *CO3* and the acetaminophen extended network on estradiol, estrone, and testosterone. Acetaminophen metabolism genes were also associated with estradiol and estrone, while their extended network was also significant for testosterone (Figure 4). Similar results were obtained for the Metabolomics workbench metabolites database, where for acetaminophen targets, we observed enriched terms for dehydroepiandrosterone and extended networks for paracetamol and acetaminophen metabolism genes were affected by estrone, estradiol, and testosterone.

## 3. Discussion

The limited number of randomized trials and the conflicting data from observational studies prevent a clear definition of the role of acetaminophen in the etiology or potential protective effect on sex-hormone-related cancers such as breast cancer, ovarian cancer, and prostate cancer. The results of studies on the impact of acetaminophen on cancer risk are varied and often contradictory. According to some authors, acetaminophen behaves as an endocrine disruptor. Endocrine disruptors (EDCs) are substances that occur naturally or are created by humans, which can interfere with the human hormonal system [97]. These substances modify or interfere with processes related to hormone production, transport, release, metabolism, and elimination in the body. EDCs can lead to adverse health effects, impacting reproductive health and developmental processes. According to some researchers, acetaminophen can be included in this group of compounds. The main impact of acetaminophen on the hormonal system manifests as its influence on reproductive health. There are reports of adverse effects observed in male offspring of mothers who used acetaminophen during pregnancy [98]. In vitro studies conducted by Kristensen et al. (2011 and 2012) suggest that both acetaminophen and acetylsalicylic acid inhibit testosterone production by fetal rat testes [46,99]. With the help of bioluminescence imaging, it has been shown that acetaminophen increases oxidative stress inside the testis of hepatocellular carcinoma reporter (HCR), which may affect mice’s fertility [100]. The results of these analyses indicate that acetaminophen can be considered an EDC.

Although studies by Lacey et al., Fairfield et al., and meta-analyses by Trabert et al. and Barnard et al. have not found a significant association between acetaminophen use and the risk of ovarian cancer [56,58,59,65], the issue remains under investigation.

However, other research, including studies by Bonovas et al. and Baandrup et al., suggests a possible protective effect [60,62], although results concerning patient survival from studies such as Nagle et al. and Dixon et al. show no clear impact on survival duration [67,68]. Evidence regarding endometrial cancer is insufficient to determine whether acetaminophen significantly affects risk. Similarly, studies on testicular cancer are inadequate to establish a definitive relationship between acetaminophen and this cancer. In prostate cancer research, results are mixed; some studies suggest a potential lower risk associated with acetaminophen use, while others, including Murad et al. and the VITamins And Lifestyle (VITAL) study, do not confirm a significant effect. For breast cancer, the evidence remains unclear, indicating a need for further research [35,37].

Analysis by Boizet-Bonhoure et al. [101] also did not provide a conclusive answer. The studies reviewed by the authors contained conflicting information, leading researchers to suggest further investigation into acetaminophen’s effects on hormonal regulation. Considering the ambiguous results and the fact that neoplastic changes in the reproductive organs are often strongly related to sex hormone levels, it is crucial to explore the relationship between hormonal changes, cancer occurrence, and acetaminophen use.

Warnings regarding the influence of acetaminophen on the hormonal balance and potential carcinogenic effects come mainly from preclinical studies, both in vitro and studies conducted with experimental animals.

Presented in the article, computational analysis of sex cancer-related phenotypes pointed out the strongest effect of acetaminophen targets on ovarian cancer and their interaction with prostate cancer.

The association between acetaminophen metabolism-related genes and breast cancer, and their network with prostate cancer, has also been confirmed. Altinoz and Korkmaz postulated that acetaminophen may reduce the risk of ovarian cancer through specific mechanisms: suppression of NF-kappa B activity, which may subsequently decrease transcription of growth factors (e.g., COX-2, VEGF, IL-8/CXCL8), which are shown to be elevated in ovarian carcinoma, and induction of characteristic reproductive atrophy due to the presence of a sex steroid-like phenolic ring [102]. Another protective element is the reduction in the glutathione pool by drug metabolite—NAPQI, which may play a role in the sterilization of premalignant ovarian lesions; inhibition of tautomerization activity of macrophage migration inhibitory factor (MIF) released from ovarian cancer, which is necessary for normal ovulation; and blockade of cytokine-induced and endothelium-derived cyclooxygenase activity [102]. In our study, as the main acetaminophen targets interacting with cancer-related genes, we identified genes that encode the inducible isoform of cyclooxygenase (COX2), which catalyzes the committed step in prostaglandin synthesis involved in inflammation and mitogenesis (gene PTGS2). One of the hypotheses assumes that genetic polymorphisms of COX-2 may reduce overall breast cancer risk or risk for its subtypes by modulating the inflammatory response. It is well known that acetaminophen is a weak inhibitor of cyclooxygenase isoforms with some selectivity toward COX2. COX-2 gene expression is stimulated by growth factors and factors involved in the inflammatory response (IL-1, TNF-α) as well as lipopolysaccharides, transcription factors, and oncogene proteins [103]. Increased expression of COX2 has been detected in many types of cancers, including endometrial cancer, prostate cancer, and invasive breast cancers [103,104]. Overexpression of COX2 is responsible for the elevated prostaglandin biosynthesis and is a characteristic feature of breast cancer. Shen et al. investigated the association between COX2 genetic variation, use of NSAIDs, and breast cancer risk, providing modest evidence that the C allele of COX2.8473 may interact with NSAIDs to reduce risk for hormone receptor-positive breast cancer. Therefore, the lack of consistent results regarding the effect of acetaminophen on the risk of cancer may be due to genetic factors [105].

Bioinformatic analysis also confirms that one of the top acetaminophen targets interacting with cancer-related genes is the sequences that encode members of the cytochrome P450 superfamily of enzymes (CYP3A4, CYP1A1, CYP1A2). They catalyze many reactions and are critically involved in biotransformation and synthesis of key endogenous substrates, e.g., cholesterol, estradiol, and arachidonic acid, as well as xenobiotics like therapeutic drugs. For example, CYP3A4 is involved in the metabolism of approximately half of the drugs used today, including acetaminophen.

CYP1A1 metabolizes several carcinogens and estrogens. Research by Kumar et al. did not confirm any correlation between prostate cancer and *CYP1A1* polymorphisms [106]. However, Xu and Tan studied the impact of single-nucleotide polymorphisms of the *CYP1A1* gene and the gene-environment interaction on the susceptibility to endometrial cancer in women (310 endometrial cancer patients and 624 healthy controls). Researchers have shown that both the rs4646421-T allele and the interaction between rs4646421 and obesity were associated with increased risk of this type of cancer [107].

According to the study of Goodman and coworkers, ovarian cancer risk may be modified by *CYP1A2* genotype (higher risk among women with the A/A genotype than among women with the C allele) and other factors that influence *CYP1A2* expression [108]. Implications of *CYP1A1* and *CYP1A2* polymorphisms (mainly single-nucleotide polymorphisms—SNPs) have been extensively studied and provide insights into the cancer pathogenesis, risk stratification, response to therapy, and potential therapeutic targets for individuals with specific *CYP1A* genotypes [109].

Next top acetaminophen targets interacting with cancer-related genes indicated by bioinformatic analysis are *CA9*—a gene encoding carbonic anhydrase IX, which is a transmembrane protein catalyzing the reversible hydration of carbon dioxide. Carbonic anhydrase is a tumor-associated, cell surface glycoprotein that participates in a variety of biological processes and shows diversity in tissue and subcellular localization. Carbonic anhydrase IX is involved in the processes of cell proliferation and transformation and is induced in response to low oxygen as a part of the hypoxic transcriptome. *CA9* is expressed in all clear-cell renal cell carcinoma but usually is not detected in most normal tissues [110,111]. It was proven that carbonic anhydrase IX is a marker of hypoxia and an adverse prognostic factor in solid tumors such as breast cancer. High expression of *CA9* is positively correlated with a higher risk of disease progression and metastases developing, independently of tumor type or site, and is a predictive marker for radio- and chemotherapy resistance [112]. CA9 inhibitors are considered a potential remedy in cancer therapy [113]. It is believed that acetaminophen may cause inhibition of mammalian isoforms I-XIV of carbonic anhydrase [114].

Using bioinformatic tools, the genes *ESR1*, *IL6*, *IL1B*, *TP53*, and *KDR* were identified as key cancer-related targets that interact with acetaminophen targets.

The *ESR1* gene encodes an estrogen receptor and regulates the transcription of many estrogen-inducible genes that play a role in growth, metabolism, and many other reproductive functions such as sexual development and gestation. The receptor encoded by the *ESR1* gene plays a pivotal role in breast and endometrial cancer [115,116,117,118]. In silico study confirms that acetaminophen has a potency to disrupt reproductive hormones by its ability to interact with reproductive hormone receptors, e.g., estrogen 1XP9, 1QKM, androgen 5CJ6, and progesterone 4OAR by hydrogen bonds, hydrophobic, and van der Waals interactions [119].

The *IL1B* and *IL6* genes encode cytokines—important mediators of the inflammatory response and maturation of B cells, as well as playing a role in a variety of cellular activities, e.g., proliferation, differentiation, and programmed cell death. The effect of recombinant human interleukin-6 (rhIL-6) on hematopoiesis, biochemical parameters, and other cytokines was evaluated in a phase I-II clinical study in 20 patients with breast cancer or non-small cell lung cancer. The study confirmed that subcutaneous injection of rhIL-6 at a dose of 10 micrograms stimulates leuko- and thrombopoiesis [120].

The *TP53* gene encodes a tumor suppressor protein containing transcriptional activation, DNA binding, and oligomerization domains. Tumor suppressor p53 maintains genome stability by regulating cell cycle arrest, apoptosis, and metabolic homeostasis. Mutations in the *p53* gene occur in almost all human tumors; however, they are not common in estrogen-responsive tumors (applies to only 20% of breast cancers, 18% of endometrial cancers, and 1.5% of cervical cancers) [121]. A study in mice showed that p53 affects the speed of oocyte development and influences the oocyte selection during oogenesis [122]. Estrogens can regulate *p53* transcription and prolong the half-life of p53 protein, which can transcriptionally regulate *ERα* and a subset of estrogen-responsive genes [121]. Hormonal activation of the p53 protein (encoded by the TP53 gene) in the mammary gland during pregnancy has been recognized as one of the most important determinants responsible for the development of latent breast cancer [123]. p53 is considered a highly relevant molecular target for therapies to reduce the risk of breast cancer [124].

The *KDR* gene—identified as a top cancer-related target interacting with acetaminophen targets—encodes one of the two receptors for VEGF, a major growth factor for endothelial cells. VEGF is the most important element affecting both physiological and pathological angiogenesis and is upregulated in many human cancers, e.g., ovarian cancer [125,126]. In experimental studies, high doses of acetaminophen increased VEGF-A levels in the fetal hepatocytes [127]. VEGF plays a critical role in liver regeneration, and higher expression of VEGF isoforms and their receptors is observed throughout liver regeneration after administration of a toxic dose of acetaminophen in rats [128].

In conclusion, the impact of acetaminophen on cancer occurrence remains difficult to define. While preclinical studies suggest a potential link to hormonal imbalances, extensive human observational data overwhelmingly support the drug’s safety, with some studies even indicating a protective effect against hormone-dependent cancers.

Assessing the impact of acetaminophen use on the risk of hormone-related cancers is challenged by several methodological limitations. The absence of randomized controlled trials with long-term follow-up necessitates reliance on observational studies, which carry significant biases. Over-the-counter availability of acetaminophen leads to underestimation of actual exposure in prescription-based registries, attenuating potential associations. Moreover, acetaminophen users often experience pain and inflammation, conditions independently linked to cancer risk, complicating causal inference. Short follow-up durations in many studies hinder the evaluation of long-term carcinogenic effects. The selection of control groups and potential publication bias further reduce data reliability.

Acetaminophen has been shown to regulate several genes and pathways related to sex-hormone-dependent cancers in both sexes, with notable interactions observed between ovarian and prostate cancers, as well as associations between drug metabolism-related genes and breast cancer. However, the limitations of bioinformatic analyses highlight the need for further experimental studies to validate or refute these in silico findings. The existing ambiguities and discrepancies underscore the importance of well-designed, larger-scale studies to better understand acetaminophen’s potential role in altering the risk of sex-hormone-related cancers.

## 4. Methods and Materials

### 4.1. Literature Review Strategy

The review on the influence of acetaminophen use on cancer risk was conducted based on the literature available in multiple electronic databases, including PubMed, Medline, Scopus, and The Cochrane Library. Publications were filtered using the following terms: acetaminophen, paracetamol, cancer, carcinomas, sex hormones, endocrine disruption, hormone-related cancer, and reproductive toxicity. The search was performed using Boolean operators (e.g., AND, OR) and MeSH terms. The search was limited to articles published in English, with no restrictions on publication date. Reference lists of relevant articles were also screened for additional studies. Both preclinical (in vitro and in vivo) and human studies were included, but emphasis was placed on critically evaluating methodological quality, population characteristics, and potential sources of bias.

Due to inconsistencies and limitations in the human data, the literature review was complemented with an exploratory bioinformatic analysis aimed at identifying possible molecular interactions between acetaminophen-related targets and genes involved in sex hormone-dependent cancers.

### 4.2. Bioinformatic Analysis

#### 4.2.1. Gene List Selection

24 acetaminophen gene targets were obtained from BindingDB (https://www.bindingdb.org, accessed on 1 September 2025). Acetaminophen metabolism–related genes were collected from MalaCards (https://www.malacards.org/card/acetaminophen_metabolism, accessed on 1 September 2025). Cancer-related genes for breast, endometrial, ovarian, prostate, and testicular cancer, as well as endometriosis, were also retrieved from the MalaCards database. All gene lists were curated to remove duplicates and irrelevant entries and integrated using official NCBI symbol ID using the NCBI_synonyms function from the Wizbionet R package v0.99.0 [129].

#### 4.2.2. Interaction Network Construction

In order to obtain first-level interactors for acetaminophen-related genes, we downloaded the entire human protein–protein interaction network from the STRING database using the StringApp v2.2.0 [130] for Cytoscape v3.10.3 [131]. Direct acetaminophen targets obtained from Binding DB were selected, followed by the extraction of their first-level cancer-related interactors. For the construction of interaction networks between cancer-related genes and acetaminophen targets, a connectivity-based ranking was applied. Cancer-related genes were ranked by their interaction with all 24 acetaminophen targets, and the top genes with the highest connectivity were selected for further visualization. Top cancer-related genes were selected based on associations with at least 5 of our cancer types of interest and interactions with acetaminophen-related genes. Network visualization was carried out using Cytoscape, with circular layouts based on decreasing node degree. Functional annotations for genes were mapped using MalaCards data, as described above.

#### 4.2.3. Enrichment Analysis

Enrichment analysis for diseases, tissues, biological pathways, and metabolites was conducted using the EnrichR API v3.2 [132]. The automated API submission for each gene list returns results as R data frames containing enriched categories, overlap genes, and *p*-values and adjusted p-values. The analysis applied a hypergeometric test with Benjamini–Hochberg correction for multiple comparisons, using the entire human genome as the background reference. Statistical significance was defined as an adjusted *p*-value ≤ 0.05.

The aim of enrichment analysis is to identify biological categories or pathways that are overrepresented among the set of genes compared with what would be expected by chance. This approach helps reveal the functional context and biological mechanisms underlying the observed molecular changes, such as disease associations, tissue specificity, or pathway involvement, providing insight into the potential physiological and pathological relevance of the analyzed genes. Ranking of significant ontological terms was performed using the Wizbionet R package based on the lowest adjusted *p*-values [129], and visualizations were generated using the ggplot2 and ggrepel R libraries. Enrichment analysis was performed for each gene list (acetaminophen targets obtained from the Binding database, first-level interactors of these targets, genes associated with drug metabolism, and first-level interactors of these genes) separately and further integrated for visualization.

Importantly, this bioinformatic analysis is exploratory and hypothesis-generating. The predicted interactions and pathway associations should be interpreted with caution, as they are based on publicly available databases and have not been validated experimentally. Future functional studies are required to confirm their biological relevance.

## 5. Conclusions

Given these barriers, current evidence remains inconclusive regarding acetaminophen’s influence on hormone-dependent cancer risk, underscoring the need for well-designed, longitudinal studies. Future research should focus on more homogeneous study populations and consider various factors related to acetaminophen use in order to provide clearer, more reliable conclusions. Advances in bioinformatics tools may also help the medical and scientific communities identify potential safety concerns associated with acetaminophen, paving the way for more definitive answers in the near future.

## Figures and Tables

**Figure 1 ijms-27-00376-f001:**
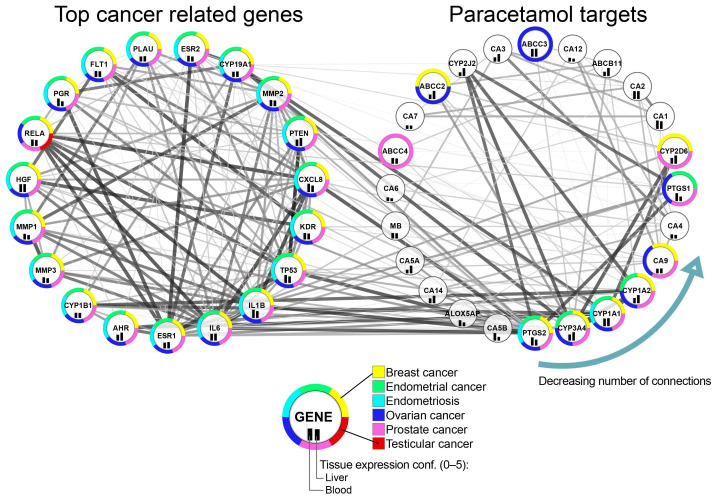
Interaction network between acetaminophen targets and top cancer-related first-level interactors. Genes were ordered circularly by decreasing the degree of connections.

**Figure 2 ijms-27-00376-f002:**
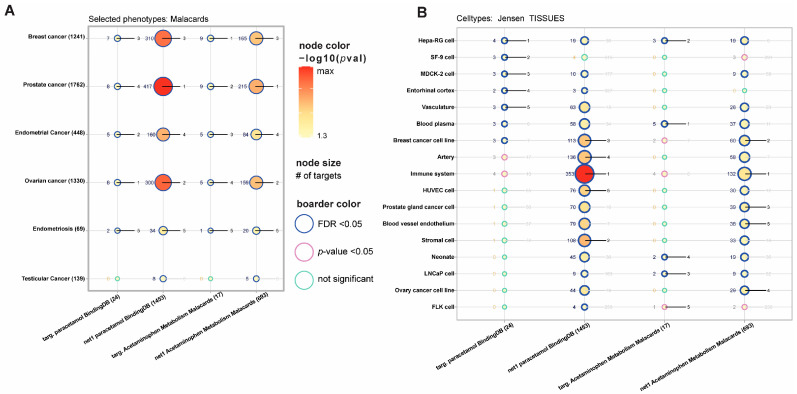
Paracetamol-related gene set enrichment analysis of (**A**) cancer-related phenotypes and (**B**) tissues. We highlighted the top five best-ranked processes. Node size is associated with enriched gene number. On the left side of the nodes is the number of enriched genes, and on the right rank (from one to five, where lower is better).

**Figure 3 ijms-27-00376-f003:**
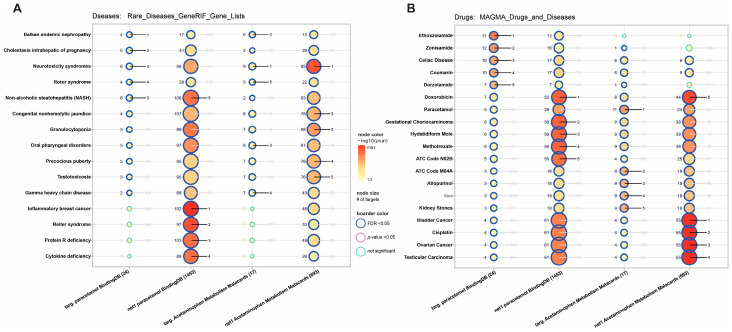
Acetaminophen-related gene set enrichment analysis of (**A**) rare disease phenotypes and (**B**) drugs and diseases. We highlighted the top five best-ranked processes. Node size is associated with enriched gene number. On the left side of the nodes is the number of enriched genes, and on the right rank (from one to five, where lower is better).

**Figure 4 ijms-27-00376-f004:**
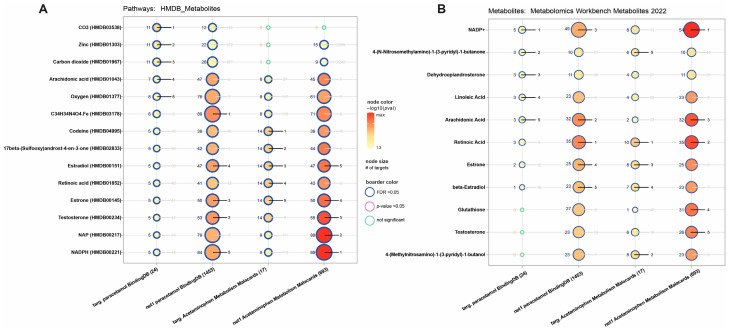
Acetaminophen-related gene set enrichment analysis of (**A**) HMDB metabolites and (**B**) metabolomic workbench metabolites. We highlighted the top five best-ranked processes. Node size is associated with enriched gene number. On the left side of the nodes is the number of enriched genes, and on the right rank (from one to five, where lower is better).

## Data Availability

Data are available from the corresponding author upon reasonable request.

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
