# Peer review of "Integrative Bioinformatic and Epidemiological Analysis of Acetaminophen Use and Risk of Sex Hormone-Related Cancers"

_ijms, 2025, doi:10.3390/ijms27010376_

Round 1

Reviewer 1 Report

Comments and Suggestions for Authors

The topic of this manuscript is very important, considering the wildspread use of paracetamol in humans. The paper is well-structured and, generally speaking, makes a good impression, as it is a source of valuable knowledge on such an important topic. It looks like the Authors have put a lot off effort into preparing this and paid attention to many aspects of possible side-effects of paracetamol use, especially in the context of hormonal problems and hormone-related cancers.

The paper, although it contains some original contribution, is predominantly a review paper and in such a case there is, in my opinion, a lot of space for Authors' flexibility in selecting and addressing particular points - some ideas could be expressed in different ways, so it is difficult to give strict guidelines on the possible revision. However, there are some fragments in this manuscript which in my opinion require correctios, as I found it exteremally difficult to understan, what the Authors have in mind:

Lines 272-274

Line 510 - what is "F interactom"?

Lines 543-544

Lines 584-586

Line 673.

Considering the overall high quality of this manuscript I think it wuild be especially important to revise all these ambiguities.

A minor point - Ref. 126 - isn't the full journal name "Acta Medica Analaya"?

An one more issue (not minor, I'm afraid) - I noticed  the complete lack of referencing and insufficient description of bioinformatic tools used in this manuscript.

Author Response

Dear Reviewer 1,

Thank you very much for your thoughtful and constructive feedback. We are pleased to hear that you recognize the importance of this topic and the overall quality of the manuscript. We have carefully considered your comments and made the necessary revisions to improve clarity and accuracy. Below, we address each of your points.

  1. Comment: The paper, although it contains some original contribution, is predominantly a review paper and in such a case there is, in my opinion, a lot of space for Authors' flexibility in selecting and addressing particular points - some ideas could be expressed in different ways, so it is difficult to give strict guidelines on the possible revision. However, there are some fragments in this manuscript which in my opinion require correctios, as I found it extremally difficult to understand, what the Authors have in mind:

Lines 272-274

Line 510 - what is "F interactom"?

Lines 543-544

Lines 584-586

Line 673.

Considering the overall high quality of this manuscript I think it would be especially important to revise all these ambiguities.

Response:

We appreciate you pointing out these ambiguous sections. We have revised these lines to improve clarity and ensure the meaning is clearer for readers. The wording has been modified where necessary to avoid misunderstandings, and we believe these changes address your concerns effectively. We agree that the term “F interactom” was unclear. This was an editorial error, which has now been corrected in the revised manuscript.

  1. Comment: A minor point - Ref. 126 - isn't the full journal name "Acta Medica Analaya"?

Response:

Thank you for pointing out the error in the journal name. We have corrected the reference to the full and accurate title, "Acta Medica Alanya."

  1. Comment: An one more issue (not minor, I'm afraid) - I noticed  the complete lack of referencing and insufficient description of bioinformatic tools used in this manuscript.

Response:

We acknowledge your concern about the lack of referencing and insufficient description of the bioinformatic tools used in the manuscript. In the revised version, we have added a detailed section describing the bioinformatic tools and methods employed in our analysis. We have also included relevant references to support our choice of tools.

We believe that these revisions have improved the clarity and quality of the manuscript, and we hope the revised version now meets your expectations. Once again, we appreciate the time and effort you dedicated to reviewing our manuscript.

Reviewer 2 Report

Comments and Suggestions for Authors

In this manuscript, the authors present a comprehensive analysis incorporating bioinformatic and epidemiological analysis to elucidate the interplay between acetaminophen and certain sex hormone-related cancers. The analysis and methodology is presented well. I would like to recommend acceptance of the manuscript following a few minor comments:

  1. How is the uniqueness determined in Fig 1 elucidating cancer related genes and paracetamol targets? Is the network connection presented the only possibility?
  2. How is enrichment in certain genes were determined and implemented in Fig 2, 3, 4?

The tables and previous work summary could be moved from the main body of the manuscript to supporting information section.

Author Response

Dear Reviewer 2,

Thank you for your valuable feedback and for recognizing the comprehensive nature of our bioinformatic and epidemiological analysis. We appreciate your suggestions and have addressed each point below:

  1. Comment: How is the uniqueness determined in Fig 1 elucidating cancer related genes and paracetamol targets? Is the network connection presented the only possibility?

Response:

Thank you for your question about the uniqueness of the network shown in Fig. 1. We clarified in the revised manuscript that the network presented is one possible interpretation based on the selection of the top multiple-cancer-related interactors, current literature and our data analysis. We acknowledge that other potential connections may exist, and we have added a note explaining that different assumptions and datasets could lead to alternative network structures.

  1. Comment: How is enrichment in certain genes were determined and implemented in Fig 2, 3, 4?

Response:

We appreciate your inquiry regarding the determination of gene enrichment in Figs. 2, 3, and 4. In response, we have added a detailed explanation of the statistical methods and bioinformatic tools used to assess gene enrichment in methods section and results. This includes the specific algorithms and criteria used to identify and evaluate the relevance of these genes in the context of acetaminophen use and hormone-related cancers.

  1. Comment: The tables and previous work summary could be moved from the main body of the manuscript to supporting information section.

Response:

In response to your suggestion, we have moved the tables and summary of previous work to the supplementary materials section. This allows for a more concise presentation in the main manuscript, focusing on the key findings and analysis.

We believe that these revisions enhance the clarity of our manuscript, and we hope the changes address your concerns. Thank you again for your insightful feedback and for helping us improve the quality of our work.